# Can Hierarchical Transformers Learn Facial Geometry?

**DOI:** 10.3390/s23020929

**Published:** 2023-01-13

**Authors:** Paul Young, Nima Ebadi, Arun Das, Mazal Bethany, Kevin Desai, Peyman Najafirad

**Affiliations:** 1Department of Computer Science, University of Texas at San Antonio, San Antonio, TX 78249, USA; 2Department of Electrical and Computer Engineering, University of Texas at San Antonio, San Antonio, TX 78249, USA; 3Department of Medicine, University of Pittsburgh, Pittsburgh, PA 15260, USA; 4Department of Information Systems, University of Texas at San Antonio, San Antonio, TX 78249, USA

**Keywords:** face geometry, hierarchical transformers, anti-spoofing, facial expression recognition, deepfakes

## Abstract

Human faces are a core part of our identity and expression, and thus, understanding facial geometry is key to capturing this information. Automated systems that seek to make use of this information must have a way of modeling facial features in a way that makes them accessible. Hierarchical, multi-level architectures have the capability of capturing the different resolutions of representation involved. In this work, we propose using a hierarchical transformer architecture as a means of capturing a robust representation of facial geometry. We further demonstrate the versatility of our approach by using this transformer as a backbone to support three facial representation problems: face anti-spoofing, facial expression representation, and deepfake detection. The combination of effective fine-grained details alongside global attention representations makes this architecture an excellent candidate for these facial representation problems. We conduct numerous experiments first showcasing the ability of our approach to address common issues in facial modeling (pose, occlusions, and background variation) and capture facial symmetry, then demonstrating its effectiveness on three supplemental tasks.

## 1. Introduction

Facial representation has always been a core part of computer vision. One of its applications, face detection, was one of the most successful image analysis tasks in the early years of computer vision [1]. Since then, facial representation applications have grown to include many security [2,3], forensics [4], and even medical [5] applications. Facial representation efforts continue to motivate innovations in the computer vision field such as federated learning advancements [6,7,8] or the introduction of angular softmax loss [9]. The face is a core element of identity and expression. People are identified primarily through their faces and communicate verbally and non-verbally through facial movements. Correspondingly, it is critical for machines attempting to interact with people to have an effective way of representing their faces.

Unexpected changes to the environment often have an adverse effect on computer vision tasks. Implementations that have vastly different background conditions from training data often generalize poorly [10]. These conditions can include variations in occlusion, lighting, background, pose, and more. Mitigating this by collecting target domain data is often costly or impractical. As a result, many efforts have been made to look for ways to mitigate these conditions without additional data. There have been more generalized attempts to address this domain shift [11] as well as specific models and methods tailored to specific tasks.

As a result of these challenges, many authors have worked to overcome these difficulties in various facial tasks [12,13,14,15,16]. Facial geometry is a representation of the physical layout of the face. This is a complex 3D structure that has some unique properties and can be simplified and represented in a variety of ways. Facial ‘keypoint’ detection is the process of detecting the location of various important parts of the face. The layout of these keypoints can be used to modify, reconstruct, or identify a face [17,18,19], whereas keypoint-based modeling can be useful for many tasks such as facial expression recognition [20], the lack of fine-grained pixel information makes it unsuitable for such tasks as spoofing or deepfake detection. Another approach uses 3D morphable models to construct a facial geometry representation, where a given face is represented by a combination of various template models [17,21,22,23]. These models can be further modified for desired deformations. However, morphable models have difficulty when encountering occluded or angled views of the face [24]. Although not fully symmetric, most faces have a certain degree of symmetry that can be exploited for facial representation tasks. This can be used to compensate for occluded information [25] or even to perform recognition with half faces [26]. We seek to capture facial geometry to create a consistent representation irrespective of these changes. Three use cases that can be used to further evaluate the capability of our facial geometry learning are face anti-spoofing, facial expression recognition, and deepfake detection.

The identity component of facial representation corresponds to face recognition and re-identification tasks. These tasks are extensively integrated into biometric security systems [27]. These systems represent an effective method of limiting access based on identity without the need or vulnerability of physical or password-based systems, whereas current face recognition methods generally operate at a high level of accuracy [28], these systems present a vulnerability to spoofing attacks. Spoofing attacks come in many forms, but the most common are photo presentation, video replay, and mask attacks. The reason for this vulnerability is that face recognition systems are trained to differentiate between different identities, not to identify false presentations. If an adversary can use a presentation attack to fool systems, the security of face-recognition-dependent systems may be compromised. The threat of these attacks has motivated many works of research into the field of face anti-spoofing (FAS); whereas the facial domain is the same, the focus for FAS shifts from a global identity to looking for subtle liveness cues such as fine-grained texture [29] or style information [12].

In addition to recognizing the person’s identity, facial representations are important for understanding the state of a person. People communicate in many more ways than just the words that are spoken. The expressions we present while interacting shape the context and meaning of the words we speak. To understand communication better, computer systems must learn to capture human emotions through expression. In addition, facial expression recognition (FER) can be used for certain medical diagnosis purposes [30,31]. The understanding of human emotion is heavily tied to multiple areas of the face: eyes, mouth, and between the eyes [32]. A facial representation that understands these expressions must be able to capture these small details while having a larger structure to localize and relate these smaller features.

Human perception can be attacked with facial manipulation techniques. This incorporates many techniques through procedures such as face swapping and expression manipulation. We will refer to this category of attacks as deepfakes. Deepfakes present a substantial threat to various facets of our lives. False criminal or civil accusations can be made with fabricated video as proof. Conversely, the utility of video evidence deteriorates if false videos cannot be detected. Similarly, elections can be swayed by conjured footage of politicians engaged in incriminating activity. Detection of these attacks can be done through the examination of regions of abnormal distortion as well as inconsistencies in the face. When facial manipulation algorithms replace or modify facial identities or expressions, there are regions that span the gap between the modified content and the original content. Deepfake algorithms are trained to make these regions as realistic as possible, but such images are still artificial, generated content with the potential for an unnatural distribution of pixels. Examining the image with sufficient resolution makes it possible to detect the artifacts left by deepfake manipulations.

We propose a deep-learning facial representation architecture that uses a multi-level transformer model to capture fine-grained and global facial characteristics. To capture the physical characteristics of the target in each sample, our proposed method uses a Swin transformer [33] which achieves state-of-the-art results in image classification tasks. The model yields facial embeddings which are used to perform various face-related tasks. Transformer architectures use attention mechanisms to represent relationships between model inputs. These architectures are effective modeling tools, but they suffer from resolution trade-offs due to their large size. The shifted window approach of our selected backbone addresses this problem allowing for better fine-grained modeling which is key for face representation tasks. Figure 1 shows a high-level representation of the proposed hierarchical architecture compared to a standard ViT transformer model. To further validate this capability, we apply this solution to three facial representation tasks: face anti-spoofing, facial expression recognition, and deepfake detection. Our performance studies show that we are able to robustly detect spoofing, deepfake attacks, and human facial expressions.

Our contributions are summarized as follows:We propose a hierarchical transformer model to learn multi-scale details of face geometry. We structure this model to capture both fine-grained details and global relationships.We validate the geometry learning capability of our facial representation approach with various tests. This demonstrates its resilience to rotation, occlusion, pose, and background variation.We apply and validate the model on three facial representation use cases. The three tasks, face anti-spoofing, facial expression recognition, and deepfake detection, highlight the robustness and versatility of facial geometry representation.

## 2. Related Work

Various early works involving facial representation used Haar-like features for face detection. The Haar sequence, proposed by Alfred Haar [34], forms the basis for Haar wavelet functions. These functions, which take the shape of a square wave over the unit interval, have been used to detect transitions such as tool failures [35]. Papageorgiou et al. and others [36,37,38] use this property on images to find boundaries for object detection. Viola and Jones [39] use a similar technique with a cascaded model for face detection. Various adaptions have been made on this such as by Mita et al. [40] which addresses correlated features and Pham et al. [41] which uses statistics for feature selection.

As convolutional neural network (CNN) models permeated computer vision, various CNN solutions for facial representation emerged [42,43,44,45]. The increased depth and number of parameters along with CNN strengths of locality and translation invariance have facilitated more sophisticated tasks such as re-identification and face recognition. The increased depth of CNNs allowed for more robust and sophisticated tasks, but tracing gradients through a large number of layers created an exploding/vanishing gradient effect. This effect limited gradient transmission to deeper layers, obstructing training and convergence of models. The introduction of residual connections [46] between layers alleviated this issue and allowed for deeper, more sophisticated ResNet architectures. However, even with this improvement, CNNs still suffer from some drawbacks. Their pooling layers can be lossy, and the convolutional architecture makes relationships between distant image regions more difficult to model.

In addition to their success in NLP, transformers have shown promising results for image processing. The capability to model image relationships with attention allows the model to focus on key regions and relationships in the image. It also better facilitates the modeling of relationships between more distant image regions. The visual transformer (ViT) [47] achieved state-of-the-art performance on image classification tasks. Due to the quadratic growth of attention parameters based on the input size, Dosovitskiy et al. structured ViT to accept relatively large patch sizes of 16×16. Various modifications have been made to utilize visual transformers for tasks such as object detection [48,49,50]. Specifically, two problems have emerged to expand the capability of visual transformers. The first is how to create different-scale feature maps to facilitate tasks such as segmentation and detection. The second is how to attend to fine-grained details without losing global relationships or overly increasing the number of parameters. The solution to both of these problems has appeared in multi-level models. Three such models [33,51,52] have appeared and each performs the following tasks in different ways. They attend to fine-grained information locally, while attending to global relationships on a coarser scale.

Structure learning problems in images require learning information from granular-level data. Transformer models provide fine-grained representation learning with attention mechanisms. The quadratic cost of attention parameters inspires many solutions to address the large size and unique challenges of images [53,54,55,56,57]. The problem distills to effectively modeling fine-grained details while preserving global relationships.

One of the simplest and most common adaptations has been the ViT model [47]. This model splits an image into 256 patches and feeds each patch as an input into a transformer encoder. The large patch sizes needed to limit the number of attention parameters deteriorate the model’s effectiveness on finer tasks such as small object detection or FAS. The Swin transformer [33] is able to shrink the size of these patches by limiting attention to local areas. It then achieves global representation by shifting the boundaries of these areas and gradually merging patches together. These smaller patch sizes make it ideal for more fine-grained tasks as it allows the model’s parameters to attend to smaller details.

Much of the recent literature on FAS has focused on building models that achieve domain generalization, attack type generalization, or both. Jia et al. [13] use an asymmetric triplet loss to group live embeddings from different domains to facilitate consistent live embeddings in new domains. Wang et al. [58] use CNN autoencoders with decoders to separate the embedding of live and spoof features to different modules. Wang et al. [12] also use CNN networks to separate style features and perform contrastive learning to suppress domain-specific features; whereas CNN architectures can capture locality information well, their global representations are limited. Similarly, certain methods such as PatchNet [29] forgo global information entirely and opt to focus anti-spoofing efforts on texture information from small patches of an image. On the other hand, there are a couple of anti-spoofing methods that make use of transformer architectures for FAS tasks. George et al. [59] use pretrained vision transformer models for zero-shot FAS capabilities. Similarly, Wang et al. [60] use a transformer model modified to capture temporal information for spoof detection. The large patch sizes of ViT [47] limit the fine-grained representation of these ViT-based models.

The challenge of facial expression recognition (FER) comes from two directions. First, models must ensure expression recognition is not tied to the identity of the individual with the expression. Second, models must learn to navigate intra-expression variance and inter-expression similarities. Zhang et al. [61] separate the identity of the face representation from the facial expression by using a ResNet-based deviation module to subtract out identity features. Ruan et al. [62] use a feature decomposition network and feature relation modules to model the relationships between expressions and mitigate the challenges related to intra-expression and inter-expression appearance. Similar to previous models, these models lack global connections and are therefore limited in their corresponding modeling ability. Xue et al. [32] use a CNN network to extract features and relevant patches and then feed them through a transformer encoder to model relations between the features and classify the image. Hong et al. [63] use a video-modified Swin transformer augmented with optical flow analysis to detect facial microexpressions. The success of this approach on that subset of expression recognition is promising for the broader use of multi-level transformer models in modeling facial expressions.

Zhao et al. [64] look at the deepfake detection problem as a fine-grained texture classification problem. They use attention with texture enhancement to improve detection. Their work sheds light on the utility of effectively modeling fine-grained image details when performing deepfake detection. This emphasis aligns with multiple other approaches which rely on fine-grained artifacts for detection [65,66,67]. In contrast, Dong et al. [68] compare identities of inner and outer face regions for their detection. These competing needs highlight the value of a combined representation of fine-grained and global relationships. There has also been some research that relates deepfake detection to expression detection. Mazaheri and Roy-Chowdhury [69] use FER to localize expression manipulation, whereas Hosler et al. [70] use discrepancies between detected audio and visual expressions to detect deepfakes. These correlations indicate that a similar model and approach may be appropriate for both problems; whereas Ilyas et al. [71] performed a combined evaluation on the audio and visual elements of a video with a Swin transformer network, it remains to be seen if the task can be performed effectively solely on image elements.

## 3. Proposed Method

Facial geometry consists of both smaller local details and larger shape and positioning. To model these well, we implement a multi-level transformer architecture that captures fine-grained details at lower layers and regional relationships at higher layers. These layers are connected hierarchically to combine these features into a complete representation. The structure of the hierarchy is determined by two factors: model depth and window size. The model depth is composed of multiple transformer blocks divided by patch merging layers as shown in Figure 2. Each transformer block consists of a multi-headed self-attention (MSA) layer and a multi-layer perceptron each with layer normalization and a skip connection. The increasing scale of these layers captures different levels of geometry features throughout the model. The window size is the number of inputs along one dimension which are attended to by each input. For example, with a window size of 7, each input would attend to a 7×7 window of inputs. In addition, the window size hyperparameter also determines the number of windows at each layer by virtue of it dividing up the number of inputs at each layer. Thus, if a layer with window size 7 receives a 56×56 input, it would contain 64 attention windows arranged in an 8×8 pattern. This change in window count affects how quickly the patch merging process achieves global attention since the number of windows is quartered at every patch merging stage. Figure 3 visualizes the feature scales using a patch size of 7.

As described by Vaswani et al. [72], MSA can be defined by the expression
(1)$$\mathrm{MSA}\left(Q,K,V\right)=\mathrm{Concat}\left(head_{1},\ldots ,head_{h}\right)W^{O}$$
where
(2)$$head_{i}=\mathrm{Attention}\left(QW_{i}^{Q},KW_{i}^{K},VW_{i}^{V}\right)$$*Q*, *K*, and *V* refer to the query, key, and value vectors derived from each input by matrix multiplication. Each *W* variable represents a different set of weights learned during training.

For each of the facial geometry tasks, the initial embeddings are created by dividing the input image into 4×4 patches. All three channels of the pixel values for these patches are concatenated into a 48-length vector. Following Dosovitskiy et al. [47], this vector is embedded linearly and given a positional encoding according to this equation:(3)$$z_{0}=\left[x_{1}E;x_{2}E;\ldots ;x_{N}E\right]+E_{pos}$$Each *x* term represents one element of the patch pixel value vector, E is the linear embedding matrix, $E_{pos}$ is the positional encoding term, and $z_{0}$ represents the final patch embedding. After the linear embedding and positional encoding, these patches are input into the transformer. The attention is localized to a M×M window of patches, where *M* is the window size. These windows are connected through a process in which window borders are shifted in subsequent layers. This places previously separated nearby patches into the same window, allowing for representation across patches, as shown in Figure 4. As the features move through the layers, these patches are gradually merged, reducing the resolution of the features while broadening the scope of the local windows. This continues until the entire representation fits within one window. Algorithm 1 breaks down this process in a step-by-step manner. Finally, the embedding is sent through a classification head according to the task required.
**Algorithm 1** Face geometry representation using hierarchical shifted windows architecture**Input:** *P* = {$p_{x}^{1},p_{x}^{2},\ldots ,p_{x}^{n}$} where $p_{x}^{n}$ is the *n*th 4×4 patch of image *x*.E = learned linear embedding matrix.$E_{pos}$ = positional encoding matrix.MSA, SW-MSA = multiheaded self-attention and shifted window MSA.MLP = multi-layer perceptronLN = layer normalization**Output:** 
Face Representation Classification 1:**for**p∈P**do**                      ▹ For each patch 2:    p← flatten(*p*) 3:    $p\leftarrow [p_{f}^{1}E;p_{f}^{2}E;\ldots ;p_{f}^{48}E]$ 4:    $p\leftarrow p+E_{pos}$ 5:**end for** 6:*X*←*P* 7:**for** block pair in transformer blocks **do** 8:    *X*← MSA(LN(*X*)) + *X* 9:    *X*← MLP(LN(*X*)) + *X*10:   *X*← SW-MSA(LN(*X*)) + *X*11:   *X*← MLP(LN(*X*)) + *X*If at block pair 1, 2, 11:12:    **for** $x_{1,1}\ldots x_{M,N}\in X$ **do**13:        $x_{m,n}\leftarrow$ merge($x_{2m,2n},x_{2m+1,2n},x_{2m,2n+1},$$x_{2m+1,2n+1}$)14:    **end for**15:**end for**

### 3.1. Use Cases

For the face anti-spoofing problem, we hypothesize that spoof-related features can be found in both the fine-grained details and the global representations. The fine-grain details include unnatural texture, reflection, or lighting. Global cues involve unnatural bowing or skew. We use our hierarchical architecture as a backbone for binary classification. The detailed representation layers give us the capability to detect based on fine cues, whereas the coarser layers enable the discovery of global spoofing cues. For training and inference, live and spoof images are sampled from their corresponding videos and classified through the model.

Similarly, deepfake detection also makes use of these diverse layers. Deepfake cues can be found both in fine-grained textures as demonstrated by Zhao et al. [64] or in larger representations, as shown by the regional identity discrepancy work of Dong et al. [68]. As with the FAS problem, we use a classification of image frames with our hierarchical architecture to detect deepfake attacks.

Facial expression recognition is somewhat different as most of the clues come from certain key regions (eyes, mouth, brow) [32]. However, these regions are not always located exactly in the same location, and thus localizing representations are needed. Furthermore, the combination of regional expressions is needed as different expressions can exhibit similar facial movements [62]. By using the aforementioned hierarchical transformer in conjunction with a multi-label classifier, we can use the various layers of features together to address these detection challenges.

### 3.2. Training

For the FAS, FER, and deepfake experiments, we fine-tuned the Swin transformer using binary and multi-class cross-entropy loss with one additional fully connected layer. Cross-entropy is an entropy-based measurement of the difference between two inputs. Specifically, it refers to the amount of information required to encode one input using the other. The cross-entropy loss for a given class *n* can be found by the equation
(4)$$l_{n}=-\sum_{c=1}^{C}w_{c}\log\frac{\exp x_{n,c}}{\sum_{i=1}^{C}\exp x_{n,i}}$$
where *x*, *y*, *w*, and *C* represent the input, target, weight, and number of classes, respectively. For binary problems, this simplifies to
(5)$$l_{n}=-w_{n}\left[y_{n}\cdot \log x_{n}+\left(1-y_{n}\right)\cdot \log\left(1-x_{n}\right)\right]$$

We selected the AdamW [73] optimizer with $\beta_{1}$ = 0.9, $\beta_{2}$ = 0.999 and weight decay = 0.01 for all training purposes.

## 4. Experiments

We test the symmetry and robustness capabilities of our facial representations with multiple experiments. The effect of pose and occlusion variance is tested by comparing the embeddings using one quantitative and one qualitative experiment. We also perform one separate experiment measuring the effect of background variation. Then, we examine the capability of our facial geometry modeling on three use cases, FAS, FER, and deepfake detection. Finally, we further explore and test the limits of the symmetry modeling capability through two occlusion-based experiments.

### 4.1. Machine Specification and Parameters

Experiments were performed on a Tesla V100-SXM2 GPU with the assistance of up to 16 Intel(R) Xeon(R) Gold 6152 CPU @ 2.10 GHz processors. For embedding experiments, and pretraining for additional experiments, we used a model pretrained on facial recognition from the FaceX-Zoo suite [74]. This locked the layer count to 2, 2, 18, 2, and the window size to 7. When training, we varied the number of frozen pretrained layers, the learning rate, and the number of training epochs. Testing was performed using train/test splits either built into the dataset (SiW) or created. Deepfake detection experiments were performed with 150 frozen layers, with a learning rate of 0.0001, and a training time of 40 epochs. Facial expression recognition was performed with 200 frozen layers, a learning rate of 0.001, and 30 training epochs. All SiW protocols were performed with a learning rate of 0.0001. Protocol 1 used 200 frozen layers and 10 epochs, protocol 2 used 200 frozen layers and 30 epochs, and protocol 3 used 200 frozen layers and 20 epochs.

### 4.2. Data Preparation

Pose and occlusion variance experiments were performed using selected images sampled from the SiW dataset [75] because of the accessibility of varied facial poses. Details of the selected frames are available in the code. Background variation experiments were performed using selected images sampled from the OULU-NPU datset [76], due to the variation in the image backgrounds.

For the use case of the facial representation capabilities of our approach, we tested our approach on three additional face-related tasks: face anti-spoofing, deepfake detection, and facial expression recognition. For the face anti-spoofing task, we used the SiW dataset [75]. SiW consists of 4478 videos of 165 subjects divided between live and spoofing videos. For the deepfake detection task, we tested the FaceForensics++ (FF++) dataset [77]. FF++ has 9431 videos consisting of 8068 attack videos as well as 1363 benign videos. The attack videos are split into five categories based on the technique used to generate them: DeepFake, Face2Face, FaceShifter, FaceSwap, and NeuralTextures. Finally, for the facial expression recognition task, we used CK+ dataset [78]. The CK+ dataset contains 593 emotion frame sequences, but we only use the 327 sequences which have labels associated with them. These images exhibit 8 emotion categories: neutral, anger, contempt, disgust, fear, happiness, sadness, and surprise. Details of these three datasets can be found in Table 1.

For testing on the video datasets, four frames were selected at random from each video. Each sequence in the CK+ dataset progresses from a neutral expression to the most expressive. The final three frames of each sequence were selected for the corresponding emotion category, and the first frame was used as the neutral category. The dataset was divided into train and test sets with a roughly 70/30 split. The frames were cropped to each subject’s face using the facial detection and cropping code in the FaceX-Zoo [74] suite. Detecting and cropping the face with this method narrows the scope of the problem to images with a single centered face image. This helps reduce the effect of multiple objects in the input image. The resulting images contained 224×224 pixels.

### 4.3. Results

To evaluate the pose and occlusion variance quantitatively, we compare the cosine similarity of four embeddings. The first embedding is one generated from an ordinary frontal image of a person. The second embedding comes from an askance image of the same person’s face. The third is from the original face with the occlusion mask placed over one eye. For comparison, we select the fourth image from a different person. The similarity among these embeddings is presented in Table 2. Note the large numbers relating images of the same person compared to that of another individual; whereas the askance embeddings show some variance from the originals, it is far less than the comparison to another individual.

For a qualitative measure of pose and occlusion variance, Figure 5 presents some examples from these occluded and askance samples. Figure 6 gives a tSNE visualization of the closeness of the intra-person embeddings for these images compared to the inter-person distances. Here you can see groupings of points representing original and alternative images of the same person labeled as the same color. The alternatives are produced either by occluding the original image (represented by the +) or selecting an image with a different pose (represented by ×). The grouping of intra-person embeddings and the separation of inter-person embeddings demonstrate the robustness of our approach to occlusion and pose.

Table 3 shows a comparison between embeddings across individuals and backgrounds. The high correlation between embeddings generated from the same person with different backgrounds shows the robustness of the proposed approach to background variation. The vastly lower correlation to embeddings from other people of the same background confirms its ability to filter out background information when performing facial geometry modeling.

Table 4 compares our results with existing works on deepfake and expression recognition. Table 5 compares anti-spoofing capability on the three protocols of the SiW dataset. Protocol 1 tests pose invariance, training on frontal views, and testing on a variety of poses. Protocol 2 performs a four-part leave-one-out strategy for the replay device. Protocol 3 tests the capability of unseen attack types by training on either print or video attacks and testing on the other.

## 5. Discussion

### 5.1. Strengths

The experiments have shown that a hierarchical transformer architecture learns a robust facial geometry representation. As shown in Figure 7, we prepare our approach to give a consistent performance with different poses and occlusions. The pose and occlusion experiments demonstrate that our model is robust against missing information and that it can extrapolate information using facial geometry representations. Similarly, the occlusion experiments demonstrate that it can use facial symmetry to work around missing information to form a consistent representation. To explore the symmetry and occlusion robustness of our approach, we performed a gradual horizontal occlusion of an image and captured the embedding outputs. Figure 8 graphs the cosine similarity between the embedding of the original and the occluded image as occlusion increases. The ability to effectively use symmetry to model facial geometry is shown by the stark contrast before and after the 50% occlusion mark. This is validated by a similar experiment performed with vertical occlusion in Figure 9. When symmetry is not present, the similarity drops far more rapidly than in the previous experiment. This shows the role that symmetry plays in accounting for missing facial geometry information.

The additional use cases further highlight the versatility of the approach. Anti-spoofing, expression recognition, and deepfake detection examine more specialized and localized regions. The demonstrated capability on these tasks in addition to the global identity representation shows the hierarchical transformer’s ability to pivot to more specialized facial representation applications without much alteration.

### 5.2. Limitations and Future Work

The third SiW protocol for the FAS task showed comparatively poor results. This protocol involves testing on unseen attack types (print vs. video). This domain generalization problem is a common and difficult problem that often requires specialized model augmentation to address it. Investigating how the hierarchical transformer can be augmented to deal with this domain generalization problem is a topic of further study.

The facial geometry modeling of this method is generated from a single image or video frame. This speeds up computation and makes the model usable with a larger variety of inputs. However, it loses the ability to capture time-related facial features such as motion. Various facial representation tasks involve motion that could be useful for classification, such as the movement of the mouth or eyes in expression recognition. To extend the functionality of this method to capture these details, our approach could be expanded to include temporal features in its decision-making. It may be worth exploring the merits of either a direct temporal expansion of the model or some augmented approach such as optical flow.

## 6. Conclusions

In this paper, we proposed a hierarchical architecture for capturing facial geometry features. This model’s ability to model both fine-grained details and global relationships makes it versatile in addressing a wide range of facial representation tasks. We demonstrated its symmetry and robust modeling capability through a series of experiments. First, we compared embeddings of various circumstances (occlusions, pose variation, and differences in the background). The consistency of the embeddings generated demonstrated the robustness of our approach to disturbances. Next, we tested symmetry modeling with a sliding window experiment. The sharp contrast between occluding less than half the face and more than half illustrated the facial symmetry modeling capabilities. Finally, we further demonstrated the flexibility of the approach by applying it to various facial representation tasks. These tasks, anti-spoofing, facial expression recognition, and deepfake detection, showcase the different ways this facial geometry modeling can be applied to problems. The results on anti-spoofing and deepfake detection showed its ability on fine-grained details whereas facial expression recognition demonstrated its ability on broader facial representation.

## Figures and Tables

**Figure 1 sensors-23-00929-f001:**
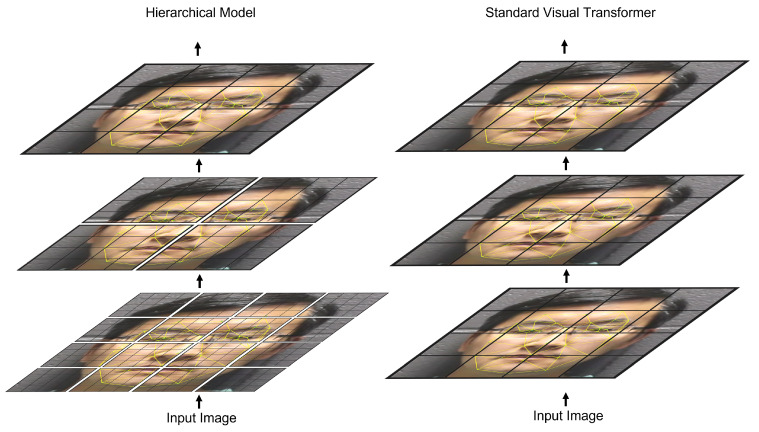
Comparison of the hierarchical model (**left**) to standard visual transformer such as ViT (**right**). In the hierarchical model, the lower layers are embedded as a greater number of smaller patches. Attention for these patches is only calculated within the local patch windows. As the image embedding progresses through the model, patches are gradually merged, and patch windows are expanded to allow for global representations.

**Figure 2 sensors-23-00929-f002:**
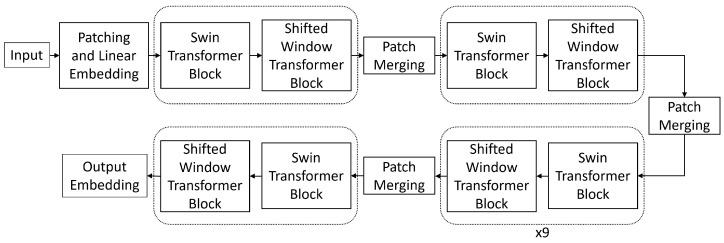
Diagram of the architecture of transformer backbone. Each transformer block has its attention parameters partitioned by windows as shown in Figure 4.

**Figure 3 sensors-23-00929-f003:**
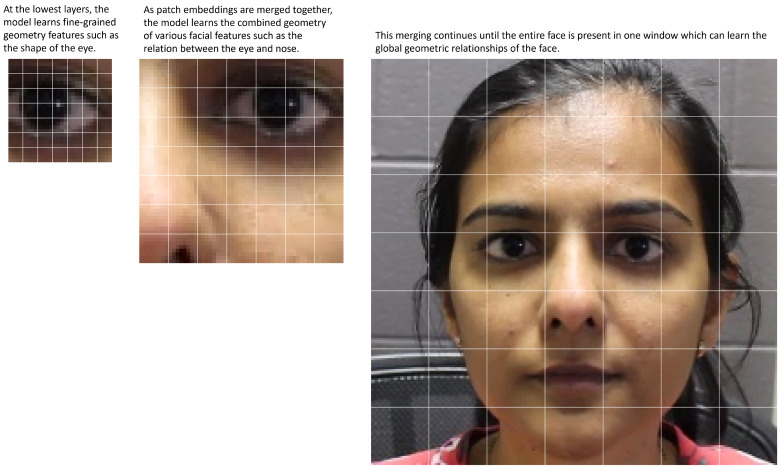
The hierarchical architecture allows the different layers to learn different parts of the facial geometry. The illustration is based on a window size of 7.

**Figure 4 sensors-23-00929-f004:**
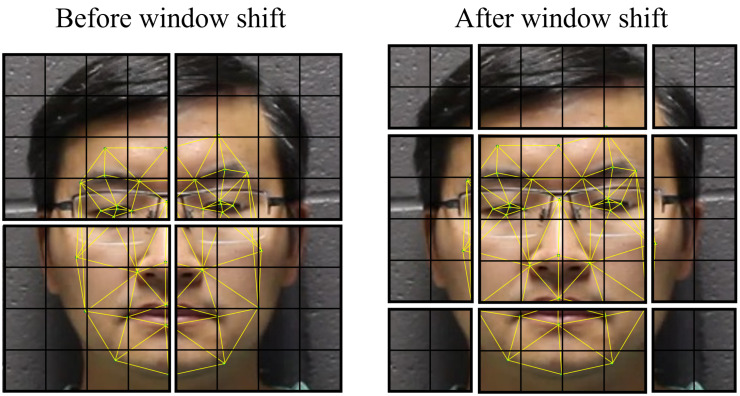
Because of localized attention windows, a mechanism is needed to represent fine-grained relationships across window borders. Window shifting provides this mechanism. By shifting the attention windows borders every other layer, we can model the relationships between each image patch and all nearby patches.

**Figure 5 sensors-23-00929-f005:**
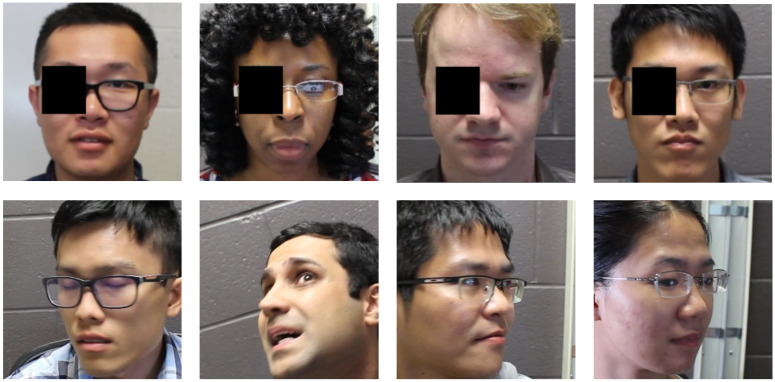
We test facial representations and symmetry capability with various modifications such as occlusions and varying facial poses. Here are examples of individuals both with eye occlusions as well as askance facial poses.

**Figure 6 sensors-23-00929-f006:**
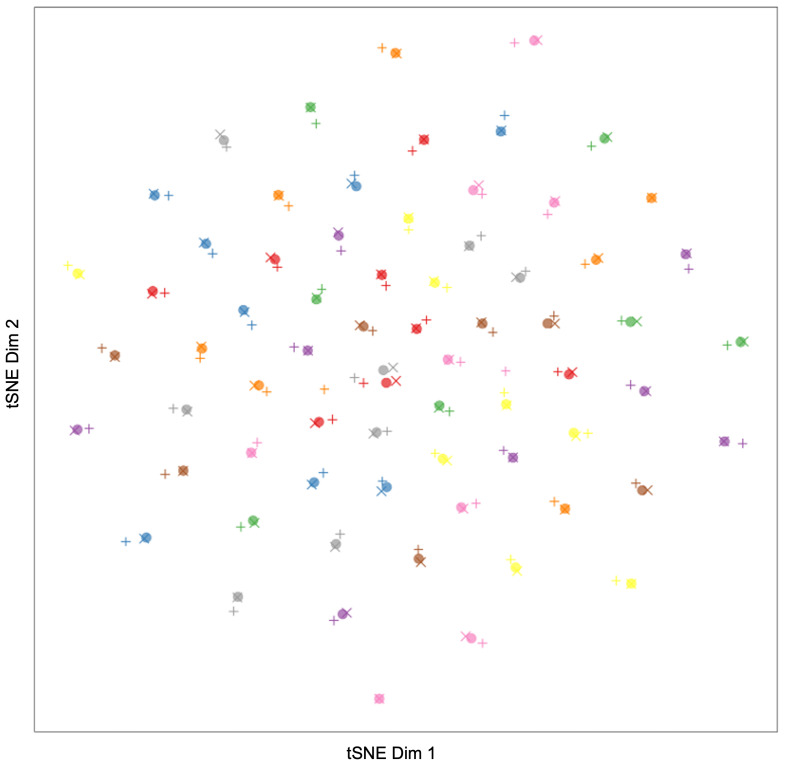
tSNE of facial embeddings from 69 individuals to show robustness to pose and occlusion. Frontal face images are represented with dots, askance images with +, and occluded images with ×. Images from the same individual are shown as the same color (with some color reuse due to a limited color palette). Note the clustering of points relating to single identities regardless of occlusion or pose.

**Figure 7 sensors-23-00929-f007:**
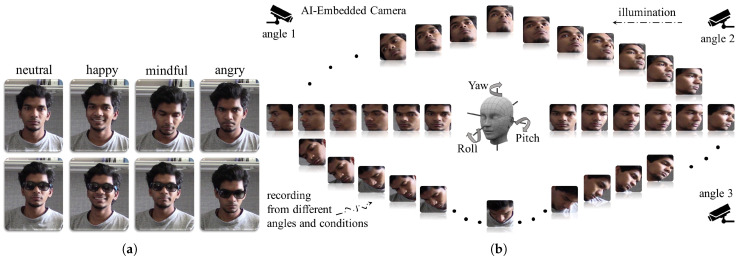
We structure and train our model to be robust to variance in pose, lighting, and occlusion. Occlusions and expressions are illustrated in (**a**), and different Yaw, Pitch, and Roll variations are illustrated in (**b**).

**Figure 8 sensors-23-00929-f008:**
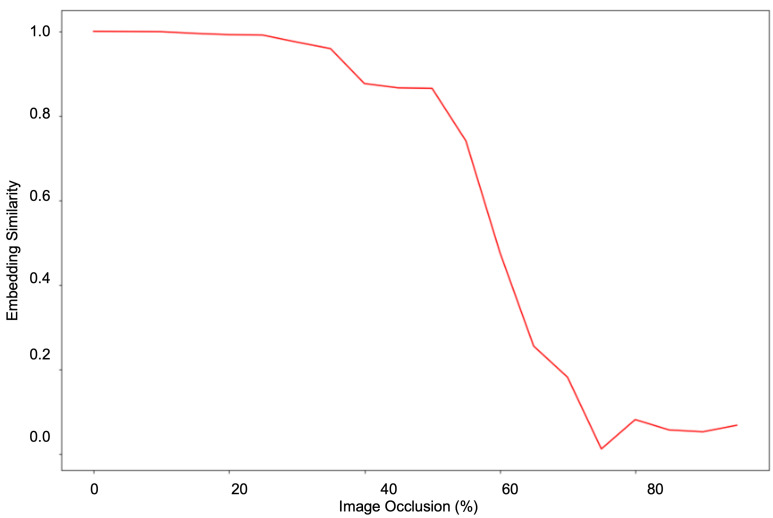
Test of the limit of occlusion robustness and symmetry capture capability. The embeddings of an original image and an occluded image are compared at various levels of horizontal occlusion. The y-axis represents the cosine similarity between the embeddings whereas the x-axis represents the percent of the image occluded.

**Figure 9 sensors-23-00929-f009:**
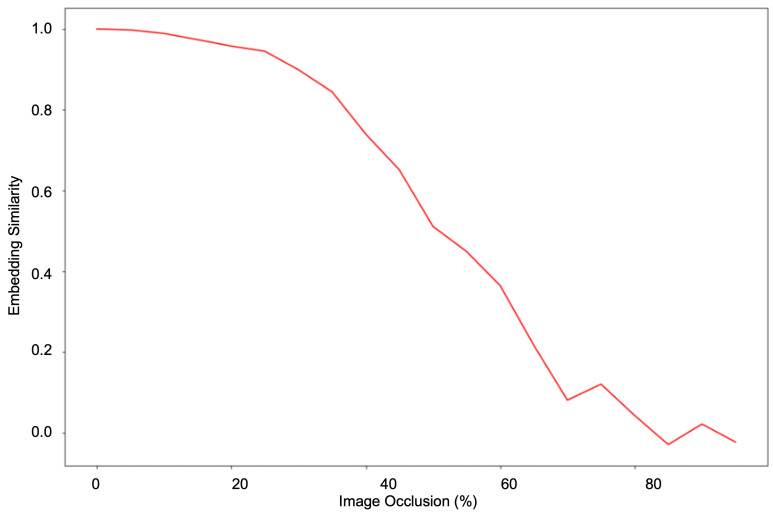
Testing gradual vertical occlusion robustness. Note the cosine similarity falls off far sooner than the horizontal occlusion experiment.

**Table 1 sensors-23-00929-t001:** Dataset descriptions: * Recorded actors only. YouTube video count not included.

Dataset	Subjects	Dataset Type	# Samples	Distribution
SiW	165	FAS	4620	1320 Live 3300 Spoof
FF++	26 *	DeepFake Detection	9431	1363 Live 8068 Manipulated
CK+	123	FER	10,735	8 Expression Categories

**Table 2 sensors-23-00929-t002:** Correlation matrix comparing average embedding distance between individuals, including occlusions and varying pose. Note the high correlation between representations of the same person and the low correlation between different people.

	Subject	Askance	Occluded	Second Subject
**Subject**	1	0.6957	0.9342	0.0995
**Askance**	0.6957	1	0.6694	0.109
**Occluded**	0.9342	0.6694	1	0.0944
**Second Subject**	0.0995	0.109	0.0944	1

**Table 3 sensors-23-00929-t003:** Testing robustness to changes in the background. Compares the cosine similarity of embeddings from each image to itself, an image from the same person with different background, an image from a different person with the same background, and an image from a different person and background. Used a selection of 60 images from 20 different actors in 3 background scenes. Each score represents the average of the comparisons taken. Images were taken from the OULU-NPU dataset [76].

	Cosine Similarity
Self	1
Different Background	0.76
Different Person	0.12
Different Person and Background	0.11

**Table 4 sensors-23-00929-t004:** Comparing AUC results (left) on FF++ [77] dataset and accuracy results (right) on CK+ [78] dataset against existing methods. Our approach shows its effectiveness in both deepfake detection and facial expression recognition.

Method	AUC	Method	Accuracy
MADD [64]	0.998	Ruan et al. [62]	0.995
Nirkin et al. [79]	0.997	PPDN [80]	0.973
Face X-ray [81]	0.984	IPA2LT [82]	0.924
Chen et al. [83]	0.984	DeRL [84]	0.974
SPSL [85]	0.969	FN2EN [86]	0.986
SMIL [87]	0.968	DDL [88]	0.992
Ours	0.943	Ours	0.957

**Table 5 sensors-23-00929-t005:** Comparison on SiW protocols for face anti-spoofing task. Protocol 1 tests anti-spoofing with unseen poses, protocol 2 tests it with varying replay mediums, and protocol 3 tests on unseen attack types (print vs. video). Our facial geometry representation is sensitive enough to transfer to fine tasks such as anti-spoofing.

Protocol	Method	APCER (%)	BPCER (%)	ACER (%)
1	ResNet(CeFA) [89]	1.03	0.83	0.93
	Yang et al. [90]	-	-	0.30
	Wang et al. [91]	0.64	0.17	0.40
	Wang et al. [58]	0.00	0.00	0.00
	PatchNet [29]	0.00	0.00	0.00
	Wang et al. [60]	0.00	0.00	0.00
	Ours	0.96	0.67	0.82
2	ResNet(CeFA) [89]	0.20±0.11	0.25±.022	0.23±0.15
	Yang et al. [90]	-	-	0.15±0.05
	Wang et al. [91]	0.00±0.00	0.04±0.08	0.02±0.04
	Wang et al. [58]	0.00±0.00	0.00±0.00	0.00±0.00
	PatchNet [29]	0.00±0.00	0.00±0.00	0.00±0.00
	Wang et al. [60]	0.00±0.00	0.00±0.00	0.00±0.00
	Ours	1.73±1.29	1.60±0.86	1.67±0.81
3	ResNet(CeFA) [89]	6.35±3.67	6.72±3.75	6.57±3.46
	Yang et al. [90]	-	-	5.85±0.85
	Wang et al. [91]	2.63±3.72	2.92±3.42	2.78±3.57
	Wang et al. [58]	4.77±5.04	2.44±2.74	3.58±3.93
	PatchNet [29]	3.06±1.10	1.83±0.83	2.45±0.45
	Wang et al. [60]	2.69±2.05	2.67±2.00	2.68±2.03
	Ours	16.81±1.66	5.03±4.24	10.92±1.29

## Data Availability

The SiW dataset can be found and used with permission at http://cvlab.cse.msu.edu/siw-spoof-in-the-wild-database.html. The CK+ dataset can be found and used with permission at http://www.jeffcohn.net/Resources. The FF++ dataset can be found and used with permission at https://github.com/ondyari/FaceForensics. Paper code is located at https://secureaiautonomylab.github.io.

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
