# Peer review of "Can Hierarchical Transformers Learn Facial Geometry?"

_sensors, 2023, doi:10.3390/s23020929_

Round 1

Reviewer 1 Report

I read this interesting article with great interest, and I wonder if similar results would be given by conducting an experiment to analyze the facial expression of, for example, a person posing sadly and a person with facial nerve palsy (in which case the face can assume the expression of a sad pose). What I mean is whether one can tell whether the face is naturally sad (because the person is sad) or rather morbidly altered (because he has a temporary dysfunction of, for example, some part of the face). Also, the Conclusions section should be completed and the Abstract section should be rewritten, as it is too generally written....

Regards

Author Response

Thank you for taking the time to review this paper and leaving detailed feedback. We have updated the document in accordance with your feedback and will address your comments below. 

Comment 1: "I wonder if similar results would be given by conducting an experiment to analyze the facial expression of, for example, a person posing sadly and a person with facial nerve palsy"
Response 1: Thank you for your suggestion. The accessibility of computer systems to people with disabilities is an important research topic and this question will be examined more in any future work on this topic.

Comment 2: "Also, the Conclusions section should be completed and the Abstract section should be rewritten, as it is too generally written"
Response 2: The abstract and conclusion sections have been expanded to be more detailed and specific.

Reviewer 2 Report

In the manuscript, the Authors have shown that the hierarchical transformer architecture proposed by them can be used to learn a robust facial geometry representation. The manuscript is well-written and easy to understand. It has extensive introduction and very solid literature review. I think that the contribution of this paper will be suitable for the MDPI Sensors. However, some minor flaws need to be corrected, as indicated in the following remarks.

·       Missing equation numbering – it makes any referencing impossible.

·       All the symbols appearing in the equations need to be explained – see, e.g., page 6. Otherwise, these equations are useless.

·     Page 6: “Following Dosovitskiy et al.,…” – missing reference number.

·     Page 7: Most style guides, academic or not, advise against introducing single subsection such as 3.1. here (without 3.2 at least).

·     Page 7: Function names such as “log” should be in roman font.

·   Page 7: The symbols x, y, w, and C below the equations should be in italic font.

·     Page 7: Undefined symbol n in the subscripts of I, x, w and y.

·    The values of beta and weight decay given here tell us nothing – they need to be commented on a bit more.

·     Table captions should be placed above them.

·  The notation of the individual items in the reference list should be made uniform. 

Author Response

Thank you for taking the time to review this paper and leaving detailed feedback. We have updated the document in accordance with your feedback and will address your comments below. 

Comment 1: "Missing equation numbering – it makes any referencing impossible."
Response 1: This has been corrected

Comment 2: "All the symbols appearing in the equations need to be explained – see, e.g., page 6. Otherwise, these equations are useless."
Response 2: All symbols appearing in equations now have explanations

Comment 3: "Page 6: “Following Dosovitskiy et al.,…” – missing reference number."
Response 3: Corrected

Comment 4: "Page 7: Most style guides, academic or not, advise against introducing single subsection such as 3.1. here (without 3.2 at least)." 
Response 4: Sections have been adjusted to avoid single subsection

Comment 5: "Page 7: Function names such as “log” should be in roman font."
Response 5: Corrected

Comment 6: "Page 7: The symbols x, y, w, and C below the equations should be in italic font."
Response 6: Corrected

Comment 7: "Page 7: Undefined symbol n in the subscripts of I, x, w and y."
Response 7: "n" is now defined to refer to the class the loss function is calculated for

Comment 8: "The values of beta and weight decay given here tell us nothing – they need to be commented on a bit more."
Response 8: These values of beta and weight decay are common values, the default for the pytorch implementation of this optimizer. No optimization or exeriments were conducted regarding them and they are included solely for reproducibility purposes.

Comment 9: "Table captions should be placed above them."
Response 9: Done

Comment 10: "The notation of the individual items in the reference list should be made uniform."
Response 10: The reference list is generated from a BibTeX file where the bibliography for nearly all papers have been obtained directly from the paper source. Hence, we are unsure how we can address this further

Reviewer 3 Report

Review: Can Hierarchical Transformers Learn Facial Geometry?

This paper explores the use of hierarquical transformers to learn facial geometry to overcome the problem of face spoofing, allowing deep fake detection.

Concerning paper organisation, it follows a general organisation, introducing slightly the problematic and and contextualisation, followed try the conducted research and experiments, evaluating several scenarios of partial information such as images patches and access the performance of the pipeline. The literature related works contextualises the problem of VIT, also presented the one of the main concept present in the pipeline, the Swin Transformers. 

The proposed architecture, explores the use of Multi-level transformers to combine fine-grained details of lower level layers with higher layers to combine regional relations, using windows to capture different image regions.

These so called multi-level Transformers, is greatly supported In the Swin Transformer scheme with a patch margin layer, and this dilutes in a large scale the novel contribution of the overall proposed work, with the novelty being the way how the window size that corresponds to the given window input is parametrised to feed the patch merging block.

The use of windows while allowing to alleviate the complexity burden, its extreme usage can reduce the way how global attention is able to capture meanfull relation among the feature space.  The window shift allows to overcome the problem of capturing/attaining visual concepts to the border patches, exploring the locality and global feature space and combine it an easy manner.  

However, an analysis of the background variation among the global image should be more explored, since the window shift can guide models to try to capture irrelevant information that will be discarded totally by the attention mechanims. At least provide some clues own the model behaves in different background, and if its robust to the changes.

For this reason, It would be interest to access how the background affects the performance of the model. Also, as suggestion in training why not use the patch random erasing to enforce a regularisation among the transformers, to better generalize to unseen images. The results should be interesting in the test dataset.

The plot from T-sne shows the performance of the proposed method, but it would be interesting to see an comparison (new plot) with images without patch or pose change, versus with to see how the distribution or spararability is affected and improved the emotion that are present in the dataset.

Overall, the hierarchy strategy is a good way to represent and be agnostic to emotions or other fake images, and capture relevant geometry features.

The T-sne plot while shows a good degree of clustering, some information such as labels that may identify or give clues t the clusters would be nice to see.

While the proposed research is coherent, however my concerns manly include:

  1.    The impact of the choice of wind size should be more deeper, with some more experiments to acmes the importance or impact of the parameter in the overall model.

  2. How image background can affect the performance of the pipeline, The results are discussed, but an study in this domain would clarify the robustness of the proposed model.

3. The abstract and intro should empathise the problematic that the proposed work tries to address. More details can be introduced to empathise the scope of the work to the reader. 

Other Issues:

  1. In some tables references, references are not referenced in text
  2. In some parts of the text, the sentences are not clear or missing some word connector, They are minor but can be fixed easily
  3. Conclusions more elaborated and discussion with a brother scenario of comparison analysis with pros and cons with more substance would enrich the conducted research.

Author Response

Thank you for taking the time to review this paper and leaving detailed feedback. We have updated the document in accordance with your feedback and will address your comments below. 

Comment 1: "The impact of the choice of wind size should be more deeper, with some more experiments to acmes the importance or impact of the parameter in the overall model."
Response 1: The focus of this paper is not a full exploration of the mechanics and optimization of the Swin transformer but the suitability and capability of these hierarchical transformers to capture facial geometry. Adding experiments testing the effect of various window sizes would sidetrack the paper into discussing the general optimization of the model rather than its connection with facial geometry. As shown in the left image of figure 4, a window size of 7 matches the size of the human eye, a key component in facial geometry. We have kept a window size of 7 for all experiments, consistent with the original paper on the Swin transformer.

Comment 2: "How image background can affect the performance of the pipeline, The results are discussed, but an study in this domain would clarify the robustness of the proposed model."
Response 2: The pose and occlusion tests were conducted using images from the SiW dataset which does not have background variation, so addressing background variation on the existing tests would not be feasible. Instead, to address this, we conducted an additional experiment testing the effect of changing background on facial embeddings. Table 3 lists the results and further discussion can be found in the experiment section.

Comment 3: "The abstract and intro should empathise the problematic that the proposed work tries to address. More details can be introduced to empathise the scope of the work to the reader."
Response 3: The abstract has been edited to provide further details and to clarify the scope and direction of the paper. This better aligns the abstract with the scope presented in the introduction.

Comment 4: "In some tables references, references are not referenced in text"
Response 4: We have added discussion with these references to correct this.

Comment 5: "In some parts of the text, the sentences are not clear or missing some word connector, They are minor but can be fixed easily"
Response 5: We have proofread the paper again and made some edits to correct sentence clarity

Comment 6: "Conclusions more elaborated and discussion with a brother scenario of comparison analysis with pros and cons with more substance would enrich the conducted research."
Response 6: Conclusion has been updated to include additional details and more discussion.

Round 2

Reviewer 3 Report

The authors have addressed some of the questions. 

However, and study with different background images would be extremely valuable.

I just have seen a text reference about it, also experiments with more than one single object should be at least discussed.

Author Response

Thank you for your review and response. We have addressed your comments below:

Comment 1: "However, and study with different background images would be extremely valuable."
Response 1: We have addressed the different background effects in Table 3 and the corresponding text.

Comment 2: "experiments with more than one single object should be at least discussed."
Response 2: Added discussion of different objects in the data preparation section (line 284)